# Red Wine Aging Techniques in Spring Water

**DOI:** 10.3390/foods14111961

**Published:** 2025-05-30

**Authors:** Danilo Rabino, Davide Allochis, Gianpiero Gerbi

**Affiliations:** 1Istituto di Scienze e Tecnologie per l’Energia e la Mobilità Sostenibili (STEMS), Consiglio Nazionale delle Ricerche (CNR), Strada delle Cacce 73, 10135 Torino, Italy; davide.allochis@stems.cnr.it; 2Independent Researcher, 14010 Cantarana, Italy; gianpiero_gerbi@yahoo.it

**Keywords:** Albugnano wine, wine maturation, wine aging, wine phenols, underwater wine

## Abstract

In wine production, technology influences its chemical composition, which in turn affects its organoleptic properties. As a result, innovative production techniques play a crucial role on the competitive wine market. Underwater wine aging has gained increasing popularity in recent years as an innovative method that can expand the variety of products available and bring engaging story telling. Some companies now offer this service to wine producers, although there is still limited knowledge about its effects on different wine types. This preliminary study investigated the impact of underwater aging by examining a well-structured red wine that was submerged for several months in spring water, comparing them to the same wine aged in a cellar for the same period. The chemical properties of the wines were analyzed after both the first (12 months of underwater and cellar aging), second (another 12 months), and third aging periods (further 12 months), to determine if there were any significant differences between them. The results revealed that underwater aging had a significant impact on the wines’ chemical composition. The dissolved oxygen level and total anthocyanin content were most notably affected by the different aging methods, while the phenolic profile and color compounds showed less influence from the treatments. The sensory test indicated that the wines aged under water and in the cellar were perceived differently, depending on the aging method and the time of evaluation (after 12, 24 or 36 months). The results of the organoleptic tests underline how the effect of the conservation environment on the sensory properties is of greater impact in the early stages of post-bottling refinement, while the differences tend to disappear when the post-bottling refinement is extended up to 36 months. The first results of a second experimental campaign seem to confirm the trends detected in the first one, although with less evidence. Further investigation is required to gain a comprehensive understanding of the complexities of underwater aging and its wider impact on wine production.

## 1. Introduction

A good red wine is characterized by a balanced structure between tannins, acidity, and alcohol, an aromatic complexity that can evolve over time, and a persistence that makes the tasting experience memorable [1].

Wine aging is a crucial process that affects its flavor profile, structure, and overall quality.

The traditional method for aging red wine is based on practices that have been consolidated over the centuries and are still used today in many wineries around the world, especially to produce high-quality wines. This method mainly involves aging in wooden barrels (mainly oak) and the use of winemaking techniques that emphasize the balance between structure, aromas, color, and complexity [2].

In recent years, several experiments have been conducted in the field of bottled wine aging conditions. Oxygen, temperature, and humidity are crucial factors that influence the aging of wine, contributing significantly to its evolution, complexity, and final quality [3]. Both must be carefully managed during aging to ensure that the wine reaches its maximum potential in terms of chemical–physical performance [4].

Firstly, temperature and humidity influence the behavior of the cork stopper [5]: it is well known that low humidity puts the structure of this material in conditions that allow for a greater flow of oxygen to the wine, while humidity between 70 and 75% allows for good hydration of the cork, such as to maintain optimal porosity, and slow and constant gas exchange. Temperature plays an important role, not so much on cork, an insulating material that is not very influenced by temperature, but rather on wine [6]. The optimal temperature for the refinement of still red wines is, in empirical terms, around 15 °C. Higher temperatures greatly accelerate the chemical dynamics, in particular the deterioration of aromas and pigments due to oxidation. Low temperatures minimize this phenomenon, but there is a limit, since low temperatures increase the wine’s ability to dissolve oxygen; furthermore, cold temperatures (below 5 °C) can block the normal evolutionary process of anthocyanin–tannin complexes, leading to a loss of color.

Following the discovery in 2010 of 168 bottles of champagne in a nineteenth century shipwreck in the Baltic Sea, some research teams in collaboration with some wineries have started testing a new method of aging wine in an underwater environment [7].

Underwater Wine is a method currently followed by around thirty producers, located both in Europe (not only in the Mediterranean area such as Italy, Spain and France, but also in the regions further north of the Baltic Sea), and in some countries bordering the Pacific and Atlantic Oceans (the United States and South America) [8,9]. Most of the companies that adopt this technique use the seabed as “submerged cellars”; however, there are also some wineries, even in Italy (Lombardy and Trentino), that exploit lake environments to preserve bottles [10].

The particular conditions in which the bottles are stored, characterized by constant temperature and pressure, absence of light and oxygen, and the movement of currents (especially in marine environments), characterize this specific refinement method.

The experience gained in recent years by various producers who have experimented with this technique highlights a particular influence of the aforementioned environmental parameters on the refinement of some wines and their chemical and organoleptic characteristics [11,12,13]. Among these studies, some results showed that underwater aging significantly influenced the chemical composition of the wines. The phenolic profile and compounds proved to be the most affected by the two different kinds of aging. The discriminant sensory test also highlighted that the underwater and cellar wines were perceived differently according to the aging treatment. However, other studies suggested that aging wine under water does not induce significant alterations in its fundamental characteristics compared to traditional cellar aging.

The study presented in this work and carried out at the Albugnano winery concerned an evaluation of the technique of refining wine under water, with a significant difference compared to the method currently adopted by various producers, concerning the depth of conservation of the wine (1–3 m). This particular difference meant that the environmental parameters that mainly influenced the maturation phase of the wine were the relatively constant temperature and the absence of light and oxygen, while the effect of depth (pressure) was marginal.

## 2. Materials and Methods

### 2.1. Study Area

The research activity described in this work was conducted in collaboration with the Azienda Agricola Orietta Perotto located in Albugnano (AT), a municipality in the Asti area located in the northern part of Monferrato, namely in the “Basso Monferrato”. This area boasts a long tradition of wine production, mainly dedicated to red wines, with the controlled designations of origin “Freisa d’Asti”, “Malvasia di Castelnuovo Don Bosco”, “Albugnano”, and the DOCG (controlled and guaranteed designation of origin) “Barbera d’Asti”. The main grape varieties grown in the area are Malvasia di Castelnuovo Don Bosco, Freisa, Bonarda, Barbera, and Nebbiolo, with which the “Albugnano” is produced (at least 85% of the grapes must be from Nebbiolo vineyards) used in this experiment.

The local climate is characterized by hot and relatively dry summers, rainy springs and autumns, and cold winters with snow events, corresponding to a transitional climate between pre-Alpine and sub-littoral [14].

The soil texture is loamy (24% clay), and the soil is classified as Typical Udorthent [15,16], derived from Miocene silty marls of the Tertiary Piedmont Basin [17]. As typical of Monferrato, the vineyards are mainly arranged with rows along the contour lines (“girapoggio”) or up and down (“rittochino”).

### 2.2. Experimental Conditions

In June 2020, a batch of “Albugnano doc Superiore” from the 2017 vintage, a wine from the Nebbiolo grape with initial aging in wooden barrels for 18 months and then in bottles for 6 months (as required by the internal regulations of “Albugnano 549” winemakers association) [18], was divided into parcels that followed a subsequent refinement in different conditions, in order to test the possibility of exploiting spring water as a natural conditioning factor. For testing, 4 theses were prepared using a pre-established number of 750 mL bottles as samples, which were prepared and placed in different storage locations.

Three theses, with 19 total samples (bottles) were placed in the winery cellar, while the fourth with 9 samples was left to refine in a well fed by spring water and located in the vineyard of the winery. The well was normally used for the vineyard water supply and was approximately 30 m deep (Figure 1, Figure 2 and Figure 3).

The selected samples for the cellar were, in turn, divided in two groups: one with 6 bottles, was refined with the conventional aging methods, while the other, consisting of 13 samples, was put for aging into a well 1 m deep located in the winery cellar and fed by spring water (Figure 4 and Figure 5).

#### 2.2.1. Preparation of Samples Placed in the Cellar and Vineyard Wells

The samples (bottles) chosen for refinement in the wells were placed under water inside different types of containers: two PVC pipes about 100 cm long with a diameter of 10 cm, hermetically sealed at both ends, each capable of holding up to 3 bottles; and a galvanized iron cage with a capacity of 6 bottles and a plastic box closed by a metal net containing a total of 10 bottles (Figure 6a–c).

Half of the samples stored in the box (5 bottles) and in the cage (3 bottles) were individually sealed inside vacuum bags, while the remaining ones were placed in the two containers directly in contact with the water from the wells. The bottles stored inside the hermetic tubes were placed inside them without any bags.

For the samples in the cellar’s well, one of the two PVC pipes and the box were used, placing the containers with the bottles on the bottom of the well at a depth of 1 m.

Otherwise, those placed in the vineyard’s well were placed inside the second PVC pipe and the galvanized cage at a depth of about 3 m, keeping the two containers suspended in the water by means of ropes fixed in correspondence with the opening of the well (Figure 7a,b).

#### 2.2.2. Monitoring Environmental Conditions: Positioning of Humidity and Temperature Probes

To monitor the environmental conditions of the two sample storage sites (cellar and vineyard), two types of probes connected to data loggers were used in order to detect and record the environmental parameters of temperature and humidity (Figure 8).

The Elitech (Elitech Ltd, London, UK) probe mod. GSP-6, equipped with a double sensor (temperature–humidity) allowed for the measurement and recording of the environmental temperature and relative humidity inside the two wells, in addition to the temperature of the water in which the samples were immersed, thanks to the positioning of one of the two sensors in the water in correspondence with the containers containing the bottles (Figure 9).

The instrument had a temperature accuracy up to ±0.5 °C (with −20° C < T < 40 °C) and ±1.0 °C (for other values) with a resolution of 0.1 °C. Humidity had an accuracy of ±3% RH (with 25 °C and 20 ÷ 80% RH) and ±5% (for other temperature and humidity conditions).

The second sensor (Elitech mod. RC-5), consisting of a single temperature probe, was used to monitor the internal environment of the cellar and was positioned near the 6 bottles intended for refinement using the traditional method (Figure 10). This instrument had the same temperature accuracy and resolution of the GSP-6 model.

The data recording interval set for the two probes was 15 min, with temperature limits between −30 °C and 60 °C. In the probe also equipped with the second sensor, the range set for the humidity values was 10 ÷ 90%.

In addition, the research team also collected climate data on temperature and relative humidity through the meteorological station of the agrometeorological network (RAM) of the Piedmont Region located in the municipality of Albugnano.

#### 2.2.3. Sample Collection and Analysis

The positioning of the samples and the temperature/humidity probes in the two conservation sites took place on 16 June 2020.

After 12 months, at the beginning of July 2021, a first inspection was carried out at the winery, when the environmental data recorded by the probes were downloaded and the initial sampling was carried out, taking a bottle from each of the containers submerged in the wells and from the group of samples refining in the cellar (control group).

The next inspection, with a new data collection from the environmental probes and a second sampling, was carried out one year later, 24 months after the start (July–August 2022). Also in this case, the same number of samples was taken from each container present in the two conservation sites.

Finally, the third and final inspection planned by the research program was carried out at the end of June 2023, at the end of the third year. On this occasion, it was not possible to collect the same number of bottles as in the previous inspections, as one of the two containers in the vineyard well, and more precisely the galvanized iron cage, was corroded and seriously damaged by being in water, with the consequent loss of the last two samples stored at the interior.

The samples from the three inspections, represented by a total of 19 bottles (cellar and vineyard), were divided as follows (Table 1):

After collection, all the samples were subjected to oenological analysis (organoleptic and chemical–physical) in the laboratory by Bi.Lab Srl (Guarene, Italy) and examined by a technical panel coordinated by the oenologist Dr. Gianpiero Gerbi for sensory evaluations.

Physicochemical parameters were assessed for different vintages. The evaluation of the analyzed samples concerned the following parameters: pH, alcoholic strength by volume (alcohol content), sulfur dioxide (free and total SO_2_), acidity (total and volatile), reducing sugars, dissolved oxygen, total polyphenols, total anthocyanins, and absorbance (420 mm, 520 mm, 620 mm). In particular, titratable acidity, pH, free sulfur dioxide, total sulfur dioxide, and ethanol data were obtained using OIV official methods [19]. Polyphenolic composition (i.e., total anthocyanins and total polyphenols) was determined as described by Di Stefano et al. [20]. More specifically, the determination of the total polyphenol content was carried out using the Folin–Ciocalteu method, measuring the absorbance of a chemically treated sample with a spectrophotometer. The identification of the total anthocyanin content was carried out using the Ribéreau–Gayon and Stonestreet method, measuring the absorbance of a chemically treated sample with a spectrophotometer. The chromatic characteristics of wines were evaluated according to OIV methods [19]. In particular, absorbance values at 420 nm, 520 nm, 620 nm were measured. Spectrophotometric analyses were carried out using a Shimadzu UV-1900 and spectrophotometer (Shimadzu, Kyoto, Japan). Each analysis was performed in triplicate.

The results obtained from the three samplings (2021, 2022, and 2023) were examined by making a comparison between the two different aging methods (traditional and underwater) between the different methods (vacuum or free) and places of conservation (cellar or vineyard) of the bottles, and taking into consideration the environmental conditions (Tables 4–6) detected with the probes placed in the two test sites.

#### 2.2.4. Organoleptic Analysis

In addition to the chemical–physical analysis, the organoleptic evaluation of the wine samples was conducted. In order to identify the differences between the different wine refinement methods, duo–trio tests were conducted on the bottles taken from the refinement site [21,22]. These tests are part of qualitative discriminant methods used in sensory analysis, according to ISO 10399:2018 and ISO 8587:2006 standards [23,24]. The purpose of this method was to identify any sensory differences between a sample and a reference. In this case, tests were conducted by comparing the “traditional aging” sample with the other samples.

More specifically, the wine samples were examined by a technical panel coordinated by the oenologist Dr. Gianpiero Gerbi. This panel was composed of 25 trained and expert judges (enologists, wine producers, trained judges) working in the wine field with a great experience of over 5 years. The rate used for grades of wine during sensory evaluation was a unipolar scale (verbal categories).

Part of our study involved the sensory analysis of a series of wine samples. As required by the applicable technical regulations, the relevant evaluation panel was selected from professional oenologists and trained judges belonging to an ISO/IEC 17025 accredited laboratory (Bi.Lab SRL). Therefore, they are not human-subjected to tests for which they require informed consent or the approval of an ethics committee, but professionals specifically trained to perform sensory evaluations.

Furthermore, the wine sampled and analyzed is a Controlled Designation of Origin wine, i.e., a product that has already obtained the approval of an institutional commission declaring it suitable for consumption and public sale.

#### 2.2.5. Following Experimental Step

In January 2024, a second batch of bottles of “Albugnano doc Superiore” from the 2020 vintage, a wine with initial aging in wooden barrels for 18 months and then in bottles for 6 months [18], was left to refine in the 1 m well fed by spring water and located in the cellar of the winery (the same one used for the first phase of the experiment). This batch consisted of 222 bottles stored in a horizontal position and enclosed in a suitable vacuum plastic bag (Figure 11). The size of the well can certainly accommodate up to 500 bottles. In the Piedmont region, many wineries and agricultural companies have a well with spring water. However, the characteristic of the individual well was not important for the purposes of our study, but rather the fact of experimenting with a technique for aging red wine in spring water.

As usual, a “control group” of bottles was refined with the conventional aging method in the cellar, in order to compare the wine characteristics during the research activity. At the same time, the monitoring of temperature and humidity parameters in the cellar and in the well continued, always by means of the same probes.

Regarding this wine, at the end of the 6-month bottle refinement period (21 February 2023), a chemical--physical analysis was carried out (Table 2).

## 3. Results and Discussion

### 3.1. Statistical Analysis of the Climatic Data

The analysis of the climatic data collected from the RAM station, and the temperature and relative humidity values recorded in the well and wine cellar highlights some interesting aspects: first, it confirms the tendency to reduce the variability of these parameters, thanks to the confined and controlled environment of the cellar, with reduced daily and seasonal fluctuations. This context promotes optimal wine aging, and the study provides scientific evidence of the achievement of favorable conditions. Figure 12 and Figure 13 highlight this trend at different time scales.

Data related to the average temperatures recorded by the probes in the cellar and in the well water inside it were analyzed and compared. A one-way ANOVA was used to compare the means of the two groups to see if there are any significant differences (*p* < 0.5). The analysis was performed using XLStat 2024.4.2 (Table 3).

To better evaluate the effect of the storage environment (cellar compared to well water) on temperatures, the periods of the year in which the values are normally more subject to significant changes were taken into consideration (hottest and coldest months). Hence, temperature data detected during June, July, August, and September for the hot periods and October, November, and December for the cold periods were considered (Figure 14 and Figure 15).

The analysis showed that the *p*-value obtained was lower than the significance level (*p* = 0.05); and therefore there was, a significant difference between the means of temperature values detected by the two probes, especially in the years 2021 and 2022, while in the first year of testing (2020), the values found were significantly different for only one month (November with a *p* = 0.007).

In the first case (2021), the months in which the differences were found were August (*p* = 0.028), November (*p* < 0.0001), and December (*p* < 0.0001). Instead, for 2022 the differences between the two groups of temperature values affected 4 months out of 7 and more precisely June (*p* = 0.013), August (*p* = 0.049), October (*p* < 0.0001), and December (*p* = 0.001).

Beyond the differences found with the statistical analysis, the values recorded with the two probes highlighted how the two types of conservation environments compared (cellar and well) were characterized by temperatures that were not very different from each other. The differences in terms of degrees were revealed to be a little more evident (by a few degrees), especially in the winter months in which the degrees measured in the water were 1–2 values higher than those in the cellar, confirming the property of water to maintain a constant temperature despite the fluctuations of the external environment.

### 3.2. Albugnano 2017: Sampling After 12 Months

The basic analytical data for the different samples were quite similar across the various tests conducted. However, some interesting findings emerged, particularly regarding dissolved oxygen levels (Figure 16). The wine sample aged using the traditional method (highlighted in red) had a dissolved oxygen content of 0.76 mg/L, which is consistent with what is typically found in “artisanal” bottling, a process carried out manually and without mechanization. This value is likely influenced by the oxygen dissolved during the bottling phase, as well as the oxygen that passes through the cork stopper [25,26,27].

The samples aged under water in the two wells (both in the cellar and in the vineyard), whether stored in vacuum or free, showed significantly different results, with much lower oxygen concentrations ranging from 0.013 to 0.079 mg/L. Humidity could have played an important role in determining the oxygen concentration in the wine, as it likely kept the cork in a more hydrated state, which would reduce oxygen permeability. Contrary to initial expectations, the absorbance data, related to the samples’ color tone, did not show significant changes. There were no marked differences between the two aging methods, with absorbance readings at 420 nm ranging from 1.8 to 1.9, at 520 nm between 1.7 and 1.8, and at 620 nm between 0.42 and 0.43.

One notable difference observed during the first maturation period (12 months) (Table 4) was in the total anthocyanin content. The concentration of these water-soluble pigments, part of the flavonoid family, was significantly higher (107 and 158 mg/L) in the vacuum-packed samples (both in the cellar and in the vineyard) compared to the traditionally aged sample (95 mg/L). Furthermore, when comparing the samples stored in tubes, both in the vineyard and cellar, the vacuum-packed samples still showed higher anthocyanin values (Figure 18).

### 3.3. Albugnano 2017: Sampling After 24 Months

The data from the samples taken after 24 months of aging (Table 5) revealed a decrease in dissolved oxygen across all the samples, further emphasizing the distinct difference between the traditionally aged wine (0.143 mg/L) and the wines aged in the two wells (vineyard and cellar), where oxygen concentrations ranged from 0.016 to 0.040 mg/L and from 0.011 to 0.021 mg/L, respectively.

When compared to the dissolved oxygen levels measured after the first 12 months, the reduction in oxygen was more significant in the traditionally aged samples during this second period. In contrast, the oxygen variation in the water-aged samples was minimal, especially in the vacuum-sealed bottles, where the levels ranged from 0.024 to 0.031 mg/L and 0.013 to 0.011 mg/L.

Regarding total anthocyanins, a clear reduction in their concentration was observed across all the samples, with very similar values regardless of the aging method or conditions. Compared to the first 12 months, the anthocyanin content in the traditionally refined samples was nearly halved, while the reduction was even more pronounced in the well-aged samples.

Another parameter that showed noticeable changes after the second maturation period (24 months) was the total polyphenol content (Figure 17), which includes compounds that influence the organoleptic qualities of the wine, such as astringency, bitterness, flavor, and color. Similar to the anthocyanin levels, the quantity of polyphenols decreased in both refinement periods, with the values being quite comparable across the different samples.

**Figure 17 foods-14-01961-f017:**
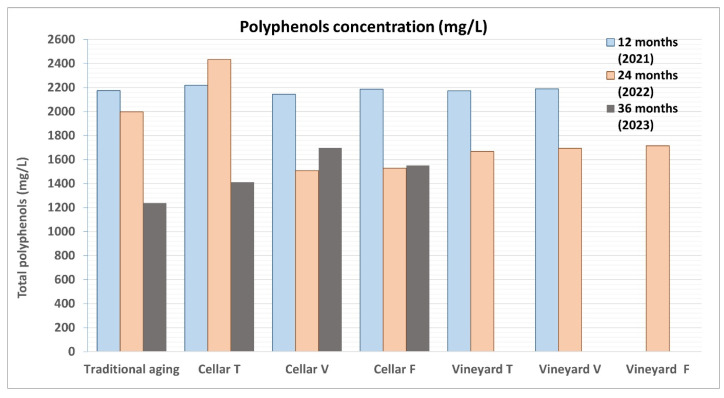
Comparison of the total polyphenol content in wine samples refined according to the two different methodologies (traditional in cellar and in vineyard and cellar wells) during 2021, 2022, and 2023. The determination of the total polyphenol content was carried out using the Folin–Ciocalteu method, measuring the absorbance of a chemically treated sample with a spectrophotometer.

**Figure 18 foods-14-01961-f018:**
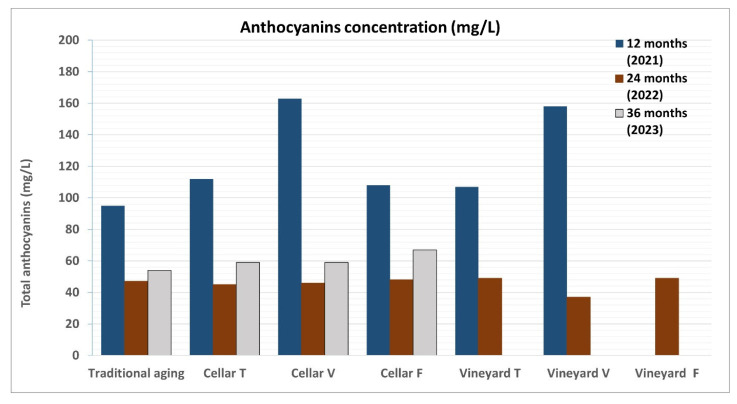
Comparison of the total anthocyanin content in wine samples refined according to the two different methodologies (traditional in cellar and in vineyard and cellar wells) during 2021, 2022, and 2023. The identification of the total anthocyanin content was carried out using the Ribéreau–Gayon and Stonestreet method, measuring the absorbance of a chemically treated sample with a spectrophotometer.

### 3.4. Albugnano 2017: Sampling After 36 Months

As the aging period was extended to 36 months (Table 6), many of the data differences diminished, particularly when comparing the two aging methods (traditional vs. water aging). For dissolved gases, the concentration was consistently higher in the traditionally aged wine, although overall, the levels in all samples had significantly decreased. This indicated a general maturation of the wine, which was also noticeable on an organoleptic level.

Precisely the parameters that show the greatest difference in the behavior of the wine between the two aging systems are those that show the least differences in the last two years of experimentation. In particular, dissolved oxygen differs after the first year by 0.739 mg/L (0.763 mg/L and 0.024 mg/L), while in the second year, it differs by 0.112 mg/L (0.143 mg/L and 0.031 mg/L), and in the third year, by 0.028 mg/L (0.042 mg/L and 0.014 mg/L).

The behavior of total anthocyanins also shows minor differences between the two aging methods in the last two years of experimentation. In more detail, they differ after the first year by 68 mg/L (95 mg/L and 163 mg/L), while in the second year, they differ by 1 mg/L (47 mg/L and 46 mg/L), and in the third year, by 5 mg/L (54 mg/L and 59 mg/L).

The most interesting finding was that total polyphenols (PTs) remained significantly higher in the samples aged in water, suggesting a more favorable outcome in terms of wine preservation. After 36 months, the color values (total anthocyanins and absorbance) showed a tendency to become more uniform across all samples.

### 3.5. Albugnano 2017: Organoleptic Analysis

Results at 12 months

All the tests gave unanimous results on the differences perceived by the examiners of the sensory panel, indicating that the “traditional aging” sample was different from the samples deriving from the refinement in water in the different methods. The most used descriptor to identify the differences with the other samples was “evolved”, indicating a greater speed of evolution of the aromas and the premature appearance of tertiary aromas in the “traditional aging” sample.

The tests were also conducted by comparing only the samples deriving from the refinement in water, identifying perceptible differences between the “free” samples and the “vacuum” samples, and identifying the difference in the intensity of the “fruit in alcohol” descriptor.

Results at 24 months

The duo–trio tests were repeated, giving the same results.

Results at 36 months

The duo–trio tests were repeated, from which the differences found were not statistically significant. In fact, the commissioners evaluated the samples differently, not identifying the “traditional aging” sample.

The results of the organoleptic tests underlined how the effect of the conservation environment on the sensory properties is of greater impact in the early stages of post-bottling refinement, while the differences tend to disappear when the post-bottling refinement is extended up to 36 months.

### 3.6. Albugnano 2020: Sampling After 12 Months

After 12 months, in February 2025, a first inspection was carried out, extracting some bottles both from those submerged in the well (2 samples) and from the control group (1 sample) (Figure 19).

The basic analytical data (Table 7) showed some interesting findings, particularly regarding dissolved oxygen levels (Figure 20). The wine sample aged using the traditional method (highlighted in red) had a dissolved oxygen content of 0.086 mg/L. The sample aged under water in the cellar well showed quite significantly different results, with a lower oxygen concentration of 0.020 mg/L. This data seem to confirm the trend observed with the Albugnano Superiore 2017 wine, although to a more limited extent. Likewise, the absorbance data, related to the samples’ color tone, did not show significant changes. There were no marked differences between the two aging methods, with absorbance readings at 420 nm ranging from 1.700 to 1.760, at 520 nm between 1.480 and 1.450, and at 620 nm at 0.210.

The content of total polyphenols does not show particularly significant behaviors, in the comparison between the two aging methods (Figure 21).

One notable difference observed during the 12-month maturation period was in the total anthocyanin content. The concentration of these water-soluble pigments was a little higher (75 mg/L) in the underwater-aged samples compared to the traditionally aged ones (68 mg/L) (Figure 22).

The organoleptic analysis highlights significant differences between underwater wine aging and the traditional one, which are characterized by a more complete, defined, and fresh olfactory profile. Also, this kind of evaluation seems to confirm the trend observed with the Albugnano Superiore 2017 wine.

## 4. Conclusions

This paper presents an initial exploration of underwater wine aging, focusing on monitoring various parameters and assessing the impact of this technique. Additionally, it is the first ever study to investigate the use of spring water for wine aging. The findings suggest that underwater aging affects wine quality differently depending on the technique used and the period of time considered.

Overall, it was observed that vacuum storage led to significant changes in some parameters (dissolved oxygen level and the total anthocyanin content), although it cannot be definitively stated that this method is superior in every aspect. Some parameters showed no significant differences between the two aging methods (underwater wine aging and traditional aging).

The fact that the best results were found in the vacuum-packed samples is also positive from a practical point of view; this way, the paper label of the bottle is avoided. Furthermore, this type of storage preserves the product from hygienic problems due to contact with spring water, and does not compromise the correct conservation of the cork stopper. Finally, the PVC pipe proved to be impractical and did not guarantee watertightness.

The limited sample size analyzed so far restricts the statistical significance of the findings. Therefore, the comparative study presented in this manuscript offers a picture of how wine responds to different aging conditions in a real-world setting.

From an organoleptic perspective, the 12-months underwater aging was positively evaluated, offering greater expressiveness and a fuller, more integrated olfactory experience. According to recent studies [13], underwater refinement demonstrated a beneficial impact, presumably due to the enhanced kinetics of reactions involving the phenolic matrix, relative to identical wines aged under conventional conditions [2,28].

The tests conducted with the second type of wine, Albugnano Superiore 2020, substantially confirm the trends identified, albeit to a lesser extent.

Lastly, an important consideration is the evaluation of the method’s impact on natural resources and energy consumption, in line with the DNSH (Do Not Significant Harm) principle. First, using groundwater does not require the addition, removal, or alteration of resources, as the bottles are simply placed in the existing water in the well, either in the cellar or vineyard, without changing its composition or state.

Additionally, unlike other methods, shallow storage in nearby sites does not require transporting bottles by truck, ship, or crane to place them in deeper waters, i.e., activities that would have a significant economic and environmental impact. Furthermore, underwater wine storage sites are often located in protected areas with sensitive ecosystems, such as marine reserves or conservation zones.

## Figures and Tables

**Figure 1 foods-14-01961-f001:**
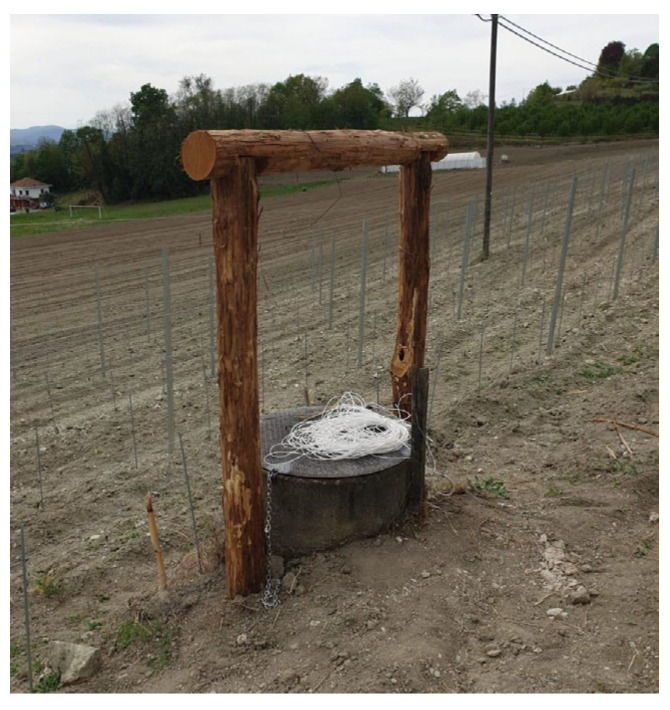
Vineyard well at the beginning of the experiment (2020), before planting the vines (rootstocks).

**Figure 2 foods-14-01961-f002:**
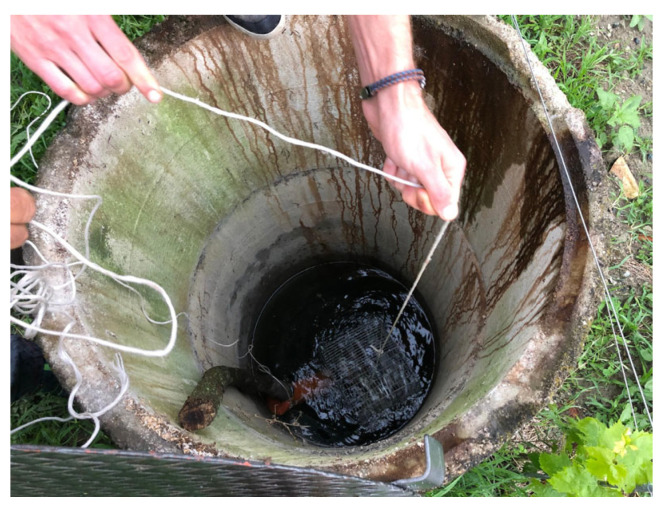
Interior of the well.

**Figure 3 foods-14-01961-f003:**
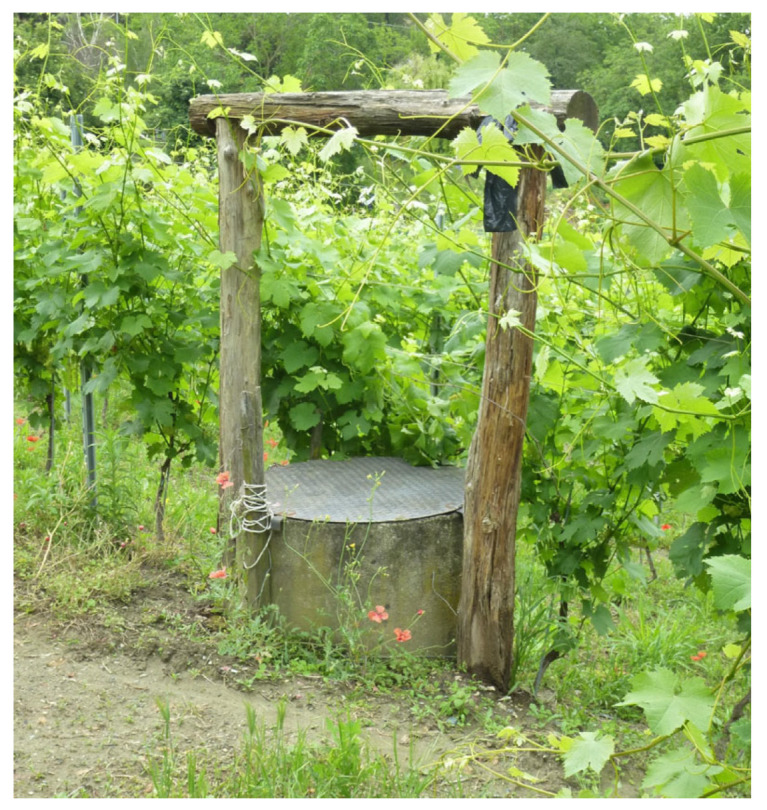
Vineyard well in 2023, during the growing season.

**Figure 4 foods-14-01961-f004:**
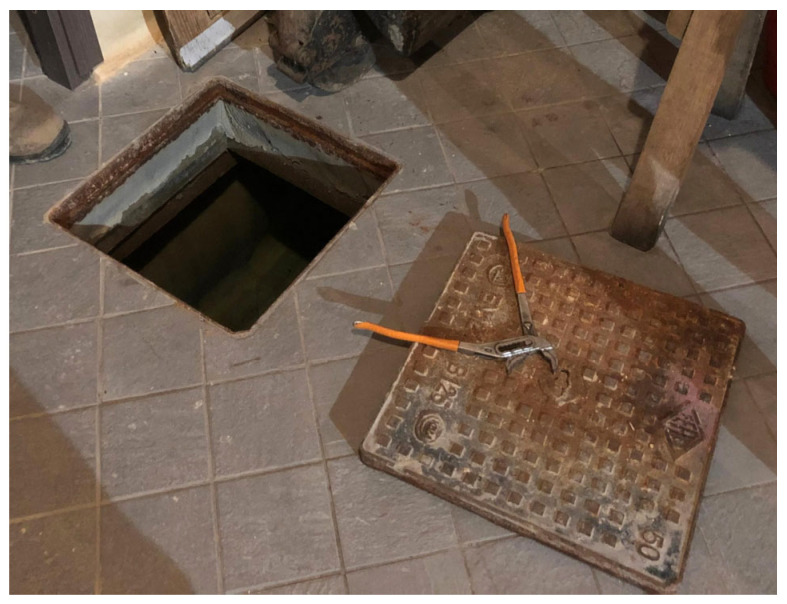
Cellar well openings.

**Figure 5 foods-14-01961-f005:**
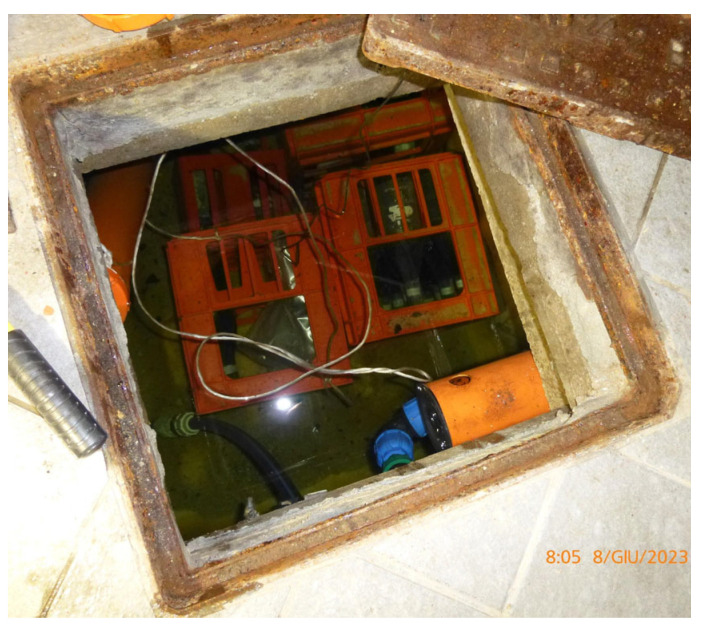
Interior of the cellar well with the samples placed to refine on the bottom.

**Figure 6 foods-14-01961-f006:**
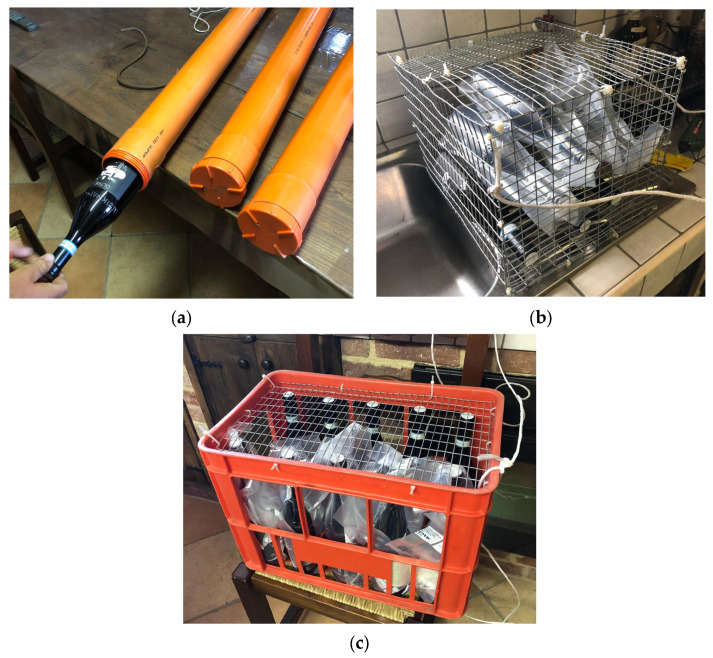
(**a**–**c**) Types of containers used for water refinement ((**a**): PVC pipes, (**b**): galvanized iron cage, and (**c**): plastic box closed by a metal mesh).

**Figure 7 foods-14-01961-f007:**
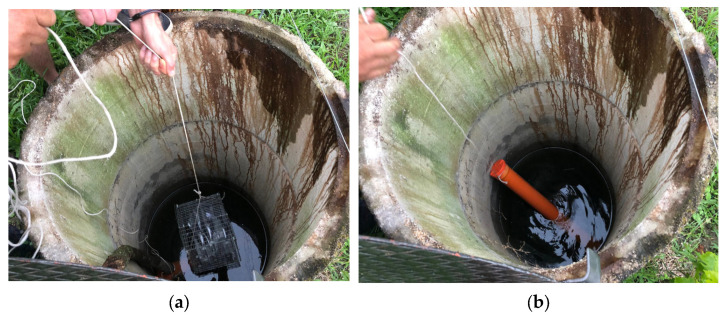
(**a**,**b**) Positioning of the cage and one of the PVC pipes inside the well in the vineyard, with the samples immersed at a depth of approximately 3 m.

**Figure 8 foods-14-01961-f008:**
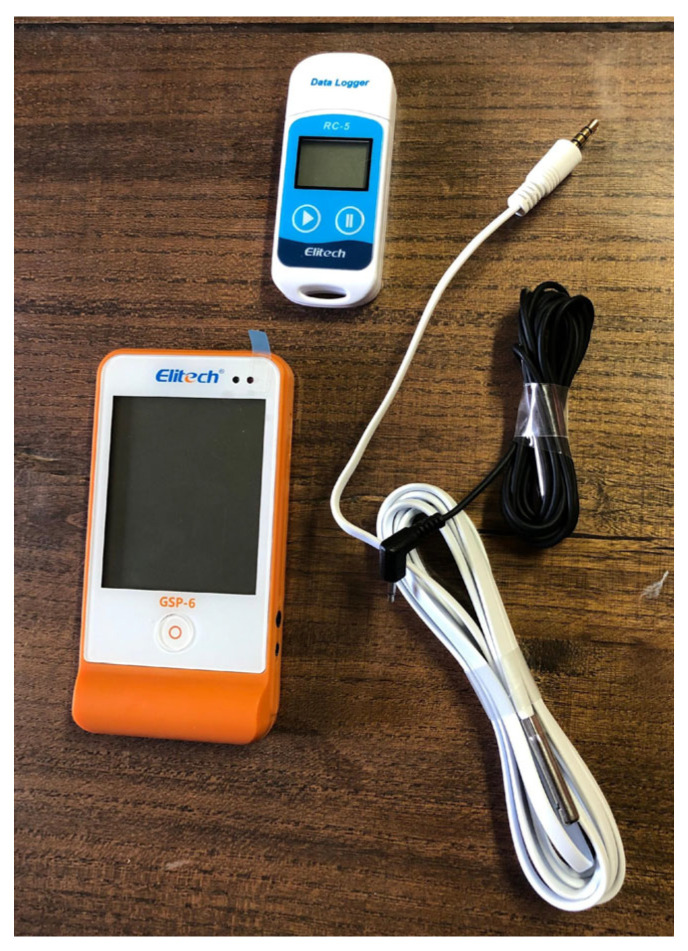
Probes used for environmental temperature and humidity detection.

**Figure 9 foods-14-01961-f009:**
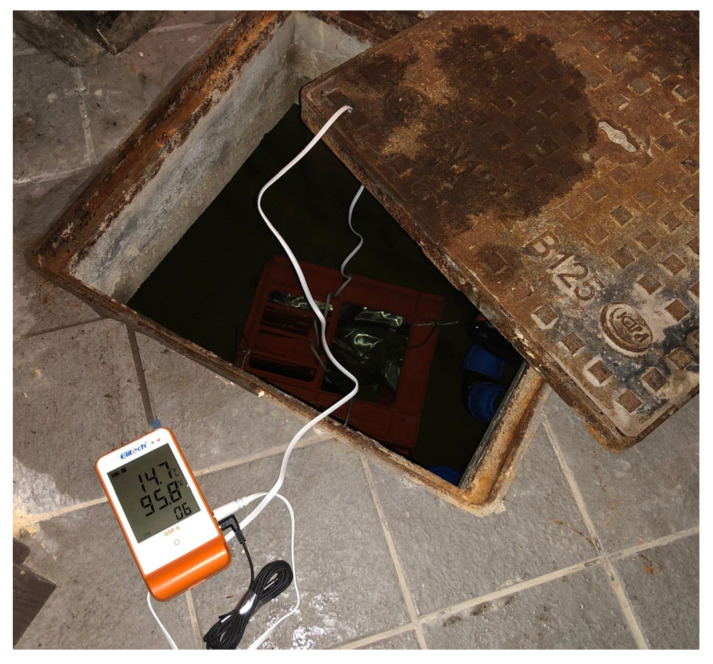
Positioning the temperature probe in correspondence with the samples in water.

**Figure 10 foods-14-01961-f010:**
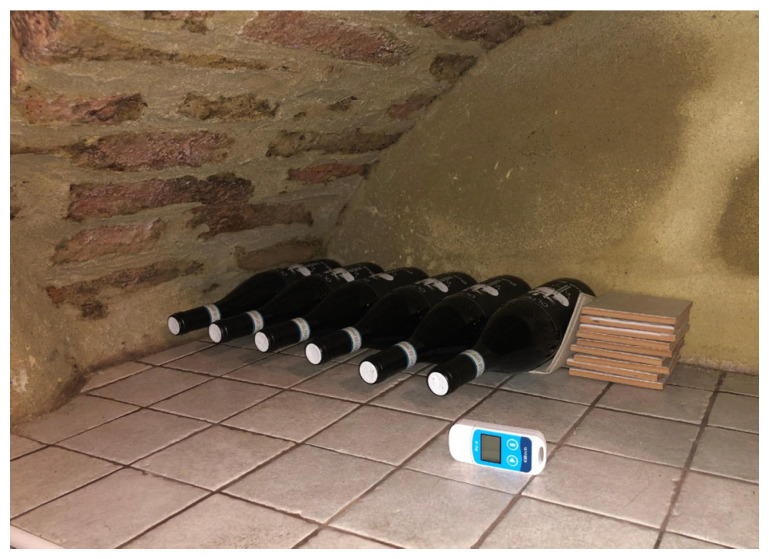
Temperature probe positioned in the cellar in correspondence with the traditionally aged samples.

**Figure 11 foods-14-01961-f011:**
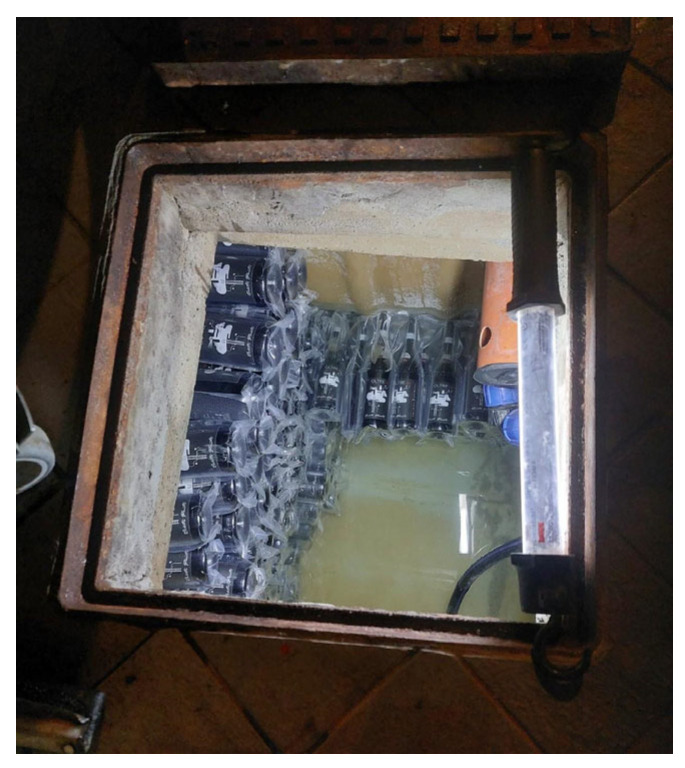
Albugnano 2020 bottles positioned in the cellar’s well.

**Figure 12 foods-14-01961-f012:**
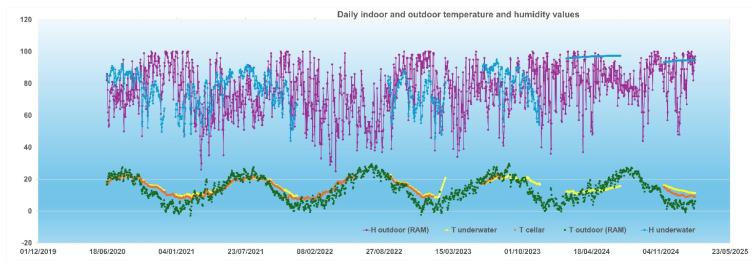
Temperature and humidity data (atmospheric and in the wine cellar) during the study period (2020 ÷ 2025).

**Figure 13 foods-14-01961-f013:**
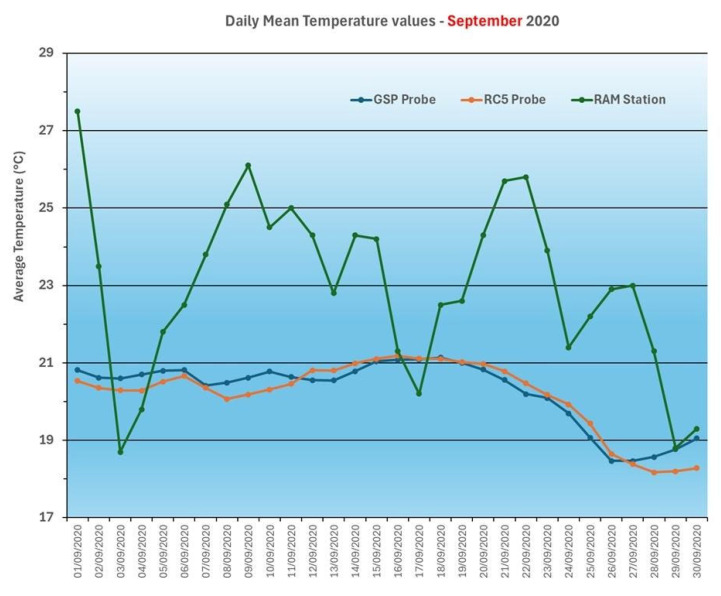
Temperature and humidity data (atmospheric by RAM station and in the wine cellar by a GSP probe for the well and RC5 probe for the cellar environment) in September 2020.

**Figure 14 foods-14-01961-f014:**
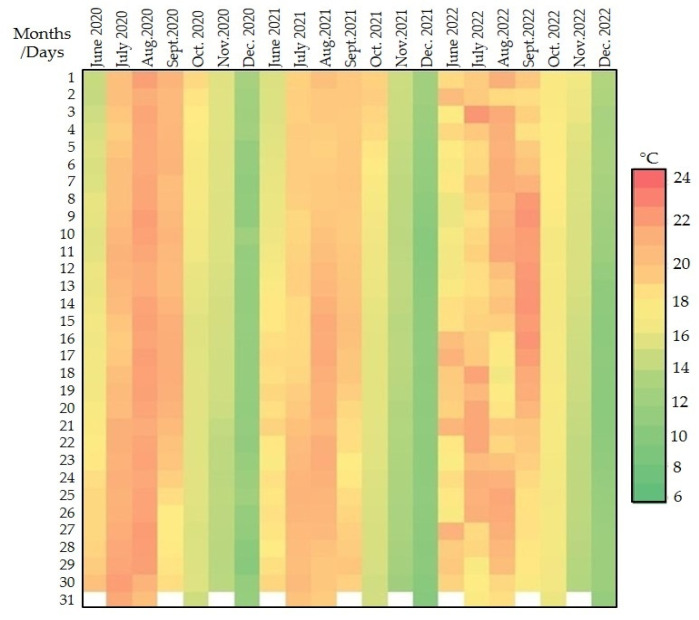
Heat map of air temperature values detected by the cellar probe.

**Figure 15 foods-14-01961-f015:**
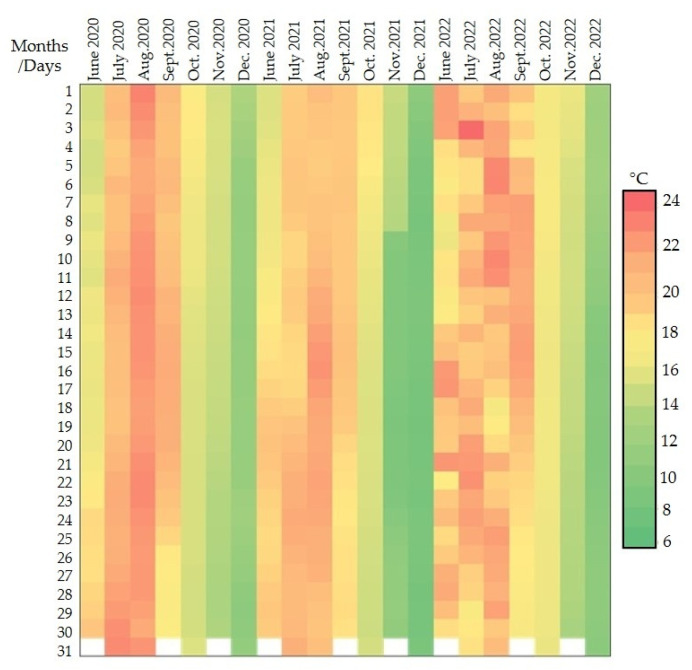
Heat map of water temperature values detected by the well’s cellar probe.

**Figure 16 foods-14-01961-f016:**
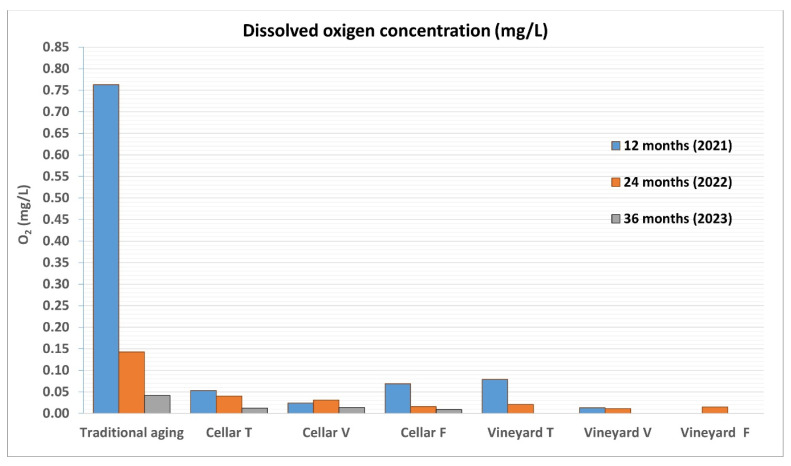
Comparison of dissolved oxygen content in wine samples (T = tube, V = vacuum, F = free) refined according to the two different methodologies (traditional in cellar and in vineyard and cellar wells) during 2021, 2022, and 2023.

**Figure 19 foods-14-01961-f019:**
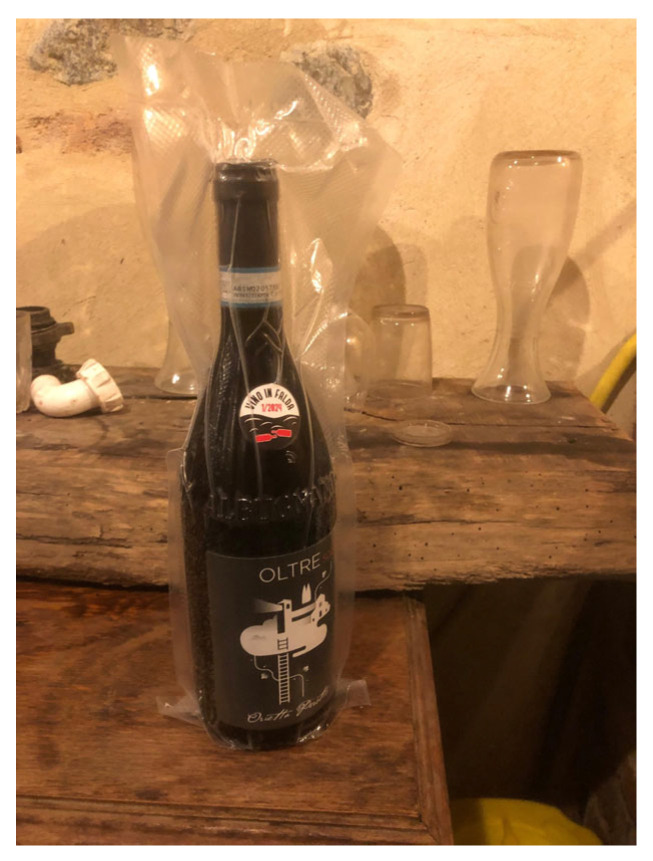
Albugnano 2020 sample extracted from cellar’s well.

**Figure 20 foods-14-01961-f020:**
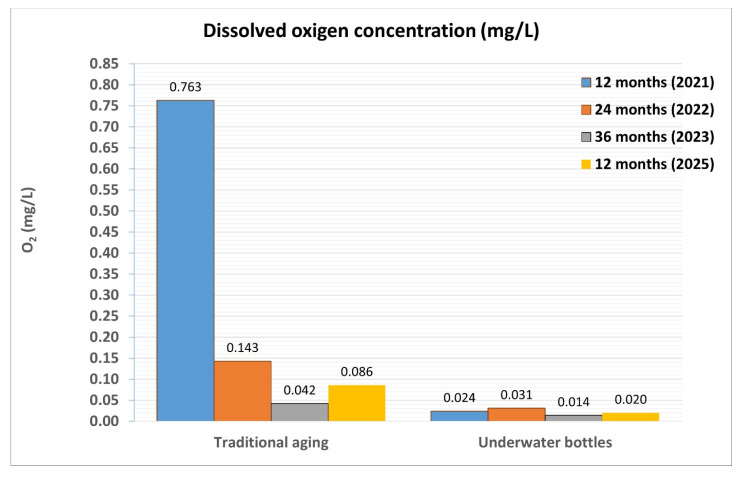
Comparison of the dissolved oxygen content in wine samples refined according to the two different methodologies (traditional and underwater) and taken during four consecutive years (2021, 2022, 2023, and 2025).

**Figure 21 foods-14-01961-f021:**
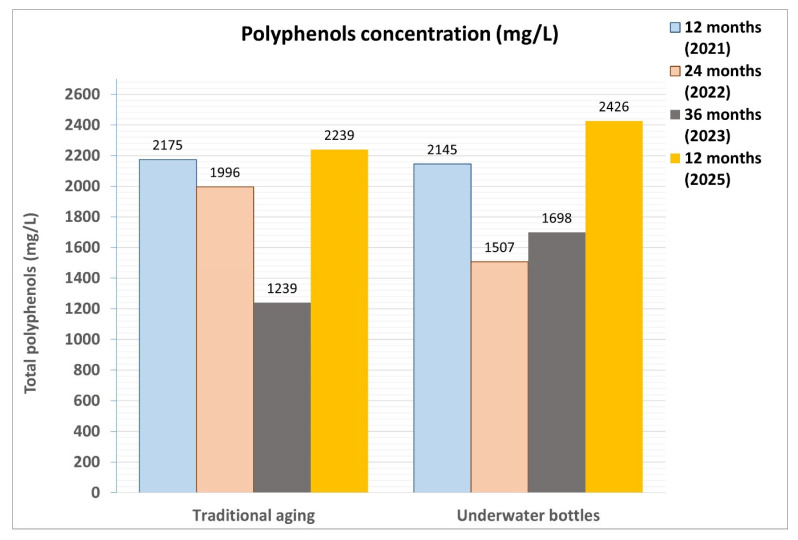
Comparison of the total polyphenol content in wine samples refined according to the two different methodologies (traditional and underwater), and taken during four consecutive years (2021, 2022, 2023, and 2025). The determination of the total polyphenol content was carried out using the Folin–Ciocalteu method, measuring the absorbance of a chemically treated sample with a spectrophotometer.

**Figure 22 foods-14-01961-f022:**
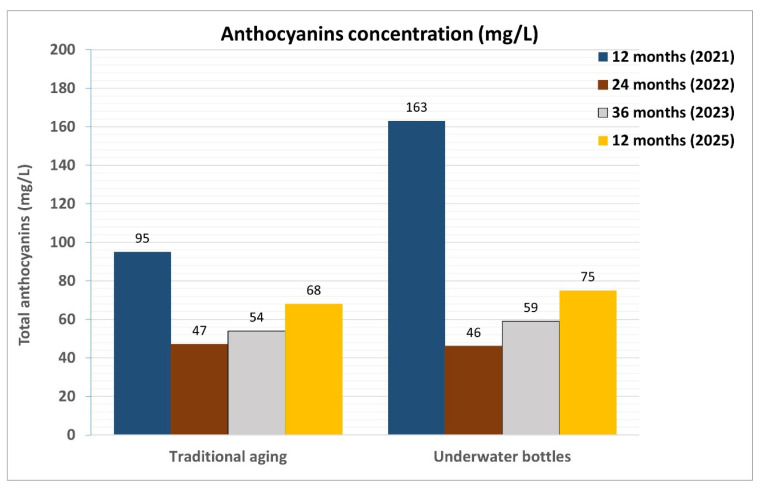
Comparison of the total anthocyanin content in wine samples refined according to the two different methodologies (traditional and underwater) and taken during four consecutive years (2021, 2022, 2023, and 2025). The identification of the total anthocyanin content was carried out using the Ribéreau–Gayon and Stonestreet method, measuring the absorbance of a chemically treated sample with a spectrophotometer.

**Table 1 foods-14-01961-t001:** Work protocol.

Conservation Site	N° of Samples	Sample Container (n.° of Bottles)	Storage Method
Cellar (well)	3	Sealed pvc tube	Free bottles
	6 (3 + 3)	Plastic box	Free bottles
			Vacuum bottles
Cellar (traditional aging)	3	No container	Free bottles
Vineyard (well)	3	Sealed pvc tube	Free bottles
	4 (2 + 2)	Iron cage	Free bottles
			Vacuum bottles

**Table 2 foods-14-01961-t002:** Chemical–physical analysis of Albugnano 2020 wine (mean ± standard deviation).

Parameters Analyzed	Albugnano Superiore 2020
Total acidity (g/L)	5.57 ± 0.01
Volatile acidity (g/L)	0.88 ± 0.02
Reducing sugars (g/L)	1.00 ± 0.01
pH	3.60 ± 0.01
Alcohol content (% *v*/*v*)	14.03 ± 0.02
Free SO_2_ (mg/L)	29 ± 1
Total SO_2_ (mg/L)	107 ± 3
Tartaric stability	32 ± 1

**Table 3 foods-14-01961-t003:** Quantitative data and ANOVA results.

Variable	Observations	Minimum	Maximum	Mean	Std. Deviation	R^2^	F	Pr > F
June 2020	60	13.90	20.29	17.25	1.61	0.00	0.02	0.89
July 2020	60	19.73	22.41	20.72	0.51	0.00	0.08	0.78
August 2020	60	20.84	22.93	21.94	0.40	0.06	3.95	0.05
September 2020	60	18.18	21.19	20.23	0.88	0.00	0.15	0.70
October 2020	60	14.90	19.25	16.57	1.13	0.06	3.91	0.05
November 2020	60	11.85	16.14	14.31	1.29	0.12	7.69	0.01
December 2020	60	9.10	11.98	10.23	0.53	0.04	2.50	0.12
June 2021	60	15.65	20.18	18.25	1.34	0.04	2.67	0.11
July 2021	60	19.20	21.22	20.01	0.57	0.04	2.51	0.12
August 2021	60	19.61	22.25	20.68	0.64	0.08	5.11	0.03
September 2021	60	18.48	20.33	19.70	0.50	0.01	0.63	0.43
October 2021	60	14.50	19.75	16.67	1.41	0.04	2.26	0.14
November 2021	60	8.20	14.58	11.47	2.15	0.44	45.33	<0.0001
December 2021	60	7.60	12.90	10.32	1.01	0.20	57.11	<0.0001
June 2022	60	16.97	22.10	19.41	1.34	0.10	6.56	0.01
July 2022	60	18.00	24.00	20.29	1.18	0.05	3.22	0.08
August 2022	60	17.50	22.80	20.53	1.21	0.07	4.04	0.05
September 2022	60	17.91	22.83	20.38	1.45	0.03	1.94	0.17
October 2022	60	17.14	18.52	17.69	0.36	0.36	33.04	<0.0001
November 2022	60	11.54	17.37	14.42	1.45	0.05	3.13	0.082
December 2022	60	8.54	12.33	10.07	0.99	0.18	13.05	0.001

**Table 4 foods-14-01961-t004:** Chemical–physical analysis of the 2021 samples (mean ± standard deviation).

Parameters Analyzed	Traditional Aging	Cellar PVC Tube	Cellar Vacuum Bottles	Cellar Free Bottles	Vineyard PVC Tube	Vineyard Vacuum Bottles	Vineyard Free Bottles
Total acidity (g/L)	6.00 ± 0.01	6.01 ± 0.01	5.99 ± 0.02	5.99 ± 0.01	6.00 ± 0.02	6.00 ± 0.01	**n.a.**
Volatile acidity (g/L)	0.80 ± 0.03	0.80 ± 0.01	0.80 ± 0.01	0.80 ± 0.01	0.81 ± 0.01	0.80 ± 0.01	**n.a.**
Reducing sugars (g/L)	1.00 ± 0.01	1.00 ± 0.01	1.00 ± 0.01	1.00 ± 0.01	1.00 ± 0.01	1.00 ± 0.01	**n.a.**
pH	3.60 ± 0.01	3.61 ± 0.01	3.60 ± 0.01	3.60 ± 0.01	3.61 ± 0.01	3.60 ± 0.01	**n.a.**
Alcohol content (% *v*/*v*)	14.380 ± 0.03	14.40 ± 0.01	14.39 ± 0.03	14.36 ± 0.04	14.32 ± 0.01	14.35 ± 0.01	**n.a.**
Free SO_2_ (mg/L)	12 ± 3	12 ± 1	11 ± 1	12 ± 3	12 ± 1	13 ± 3	**n.a.**
Total SO_2_ (mg/L)	58 ± 1	59 ± 1	58 ± 1	63 ± 1	65 ± 1	66 ± 1	**n.a.**
CO_2_ (g/L)	0.230 ± 0.02	0.210 ± 0.01	0.190 ± 0.01	0.230 ± 0.02	0.220 ± 0.01	0.220 ± 0.01	**n.a.**
Dissolved O_2_ (mg/L)	0.763 ± 0.01	0.053 ± 0.01	0.024 ± 0.02	0.069 ± 0.01	0.079 ± 0.03	0.013 ± 0.01	**n.a.**
Total polyphenols (mg/L)	2175 ± 3	2219 ± 3	2145 ± 1	2187 ± 1	2173 ± 2	2189 ± 4	**n.a.**
Total anthocyanins (mg/L)	95 ± 2	112 ± 2	163 ± 1	108 ± 3	107 ± 1	158 ± 6	**n.a.**
420 nm absorbance	1.911 ± 0.04	1.869 ± 0.01	1.902 ± 0.03	1.863 ± 0.01	1.845 ± 0.01	1.848 ± 0.01	**n.a.**
520 nm absorbance	1.537 ± 0.01	1.499 ± 0.02	1.519 ± 0.03	1.501 ± 0.02	1.512 ± 0.02	1.491 ± 0.02	**n.a.**
620 nm absorbance	0.342 ± 0.02	0.332 ± 0.01	0.336 ± 0.02	0.330 ± 0.01	0.333 ± 0.01	0.329 ± 0.01	**n.a.**

**Table 5 foods-14-01961-t005:** Chemical–physical analysis of 2022 samples (mean ± standard deviation).

Parameters Analyzed	Traditional Aging	Cellar PVC Tube	Cellar Vacuum Bottles	Cellar Free Bottles	Vineyard PVC Tube	Vineyard Vacuum Bottles	Vineyard Free Bottles
Total acidity (g/L)	6.13 ± 0.01	6.13 ± 0.01	6.12 ± 0.01	6.11 ± 0.01	6.11 ± 0.01	6.15 ± 0.03	6.08 ± 0.01
Volatile acidity (g/L)	0.79 ± 0.01	0.78 ± 0.02	0.78 ± 0.01	0.78 ± 0.01	0.78 ± 0.02	0.78 ± 0.01	0.79 ± 0.01
Reducing sugars (g/L)	1.00 ± 0.01	1.00 ± 0.01	1.00 ± 0.01	1.00 ± 0.01	1.00 ± 0.01	1.00 ± 0.01	1.00 ± 0.01
pH	3.62 ± 0.01	3.62 ± 0.01	3.62 ± 0.01	3.62 ± 0.01	3.62 ± 0.01	3.62 ± 0.01	3.63 ± 0.01
Alcohol content (% *v*/*v*)	14.40 ± 0.01	14.39 ± 0.01	14.38 ± 0.01	14.42 ± 0.01	14.40 ± 0.01	14.39 ± 0.01	14.39 ± 0.01
Free SO_2_ (mg/L)	10 ± 2	9 ± 1	11 ± 2	12 ± 2	9 ± 4	12 ± 1	10 ± 1
Total SO_2_ (mg/L)	52 ± 4	56 ± 2	60 ± 2	5 ± 1	48 ± 4	59 ± 2	60 ± 4
CO_2_ (g/L)	0.230 ± 0.02	0.220 ± 0.01	0.210 ± 0.01	0.210 ± 0.01	0.190 ± 0.02	0.200 ± 0.01	0.210 ± 0.01
Dissolved O_2_ (mg/L)	0.143 ± 0.02	0.040 ± 0.01	0.031 ± 0.02	0.016 ± 0.01	0.021 ± 0.04	0.011 ± 0.01	0.015 ± 0.01
Total polyphenols (mg/L)	1996 ± 2	2433 ± 4	1507 ± 2	1528 ± 8	1667 ± 4	1694 ± 2	1715 ± 4
Total anthocyanins (mg/L)	47 ± 4	45 ± 4	46 ± 2	48 ± 2	49 ± 3	37 ± 2	49 ± 3
420 nm absorbance	2.325 ± 0.01	2.335 ± 0.02	2.240 ± 0.01	2.245 ± 0.03	2.390 ± 0.02	2.230 ± 0.02	2.235 ± 0.01
520 nm absorbance	1.805 ± 0.01	1.830 ± 0.02	1.785 ± 0.01	1.780 ± 0.02	1.845 ± 0.01	1.760 ± 0.04	1.780 ± 0.01
620 nm absorbance	0.405 ± 0.02	0.440 ± 0.01	0.420 ± 0.02	0.415 ± 0.02	0.440 ± 0.01	0.415 ± 0.02	0.405 ± 0.02

**Table 6 foods-14-01961-t006:** Chemical–physical analysis of 2023 samples (mean ± standard deviation).

Parameters Analyzed	Traditional Aging	Cellar PVC Tube	Cellar Vacuum Bottles	Cellar Free Bottles	Vineyard PVC Tube	Vineyard Vacuum Bottles	Vineyard Free Bottles
Total acidity (g/L)	6.08 ± 0.02	6.07 ± 0.01	6.06 ± 0.03	6.05 ± 0.01	n.a.	n.a.	n.a.
Volatile acidity (g/L)	0.78 ± 0.01	0.78 ± 0.02	0.78 ± 0.01	0.78 ± 0.02	n.a.	n.a.	n.a.
Reducing sugars (g/L)	1.00 ± 0.02	1.00 ± 0.01	1.00 ± 0.02	1.00 ± 0.01	n.a.	n.a.	n.a.
pH	3.70 ± 0.01	3.70 ± 0.01	3.69 ± 0.01	3.70 ± 0.02	n.a.	n.a.	n.a.
Alcohol content (% *v*/*v*)	14.44 ± 0.01	14.43 ± 0.02	14.42 ± 0.01	14.38 ± 0.02	n.a.	n.a.	n.a.
Free SO_2_ (mg/L)	5 ± 1	7 ± 1	8 ± 2	5 ± 1	n.a.	n.a.	n.a.
Total SO_2_ (mg/L)	48 ± 2	42 ± 3	37 ± 2	29 ± 1	n.a.	n.a.	n.a.
CO_2_ (g/L)	0.190 ± 0.01	0.210 ± 0.02	0.230 ± 0.02	0.230 ± 0.01	n.a.	n.a.	n.a.
Dissolved O_2_ (mg/L)	0.042 ± 0.01	0.012 ± 0.01	0.014 ± 0.01	0.009 ± 0.02	n.a.	n.a.	n.a.
Total polyphenols (mg/L)	1239 ± 4	1412 ± 4	1698 ± 4	1551 ± 2	n.a.	n.a.	n.a.
Total anthocyanins (mg/L)	54 ± 2	59 ± 1	59 ± 2	67 ± 2	n.a.	n.a.	n.a.
420 nm absorbance	2.410 ± 0.01	2.300 ± 0.02	2.240 ± 0.02	2.340 ± 0.01	n.a.	n.a.	n.a.
520 nm absorbance	1.840 ± 0.04	1.780 ± 0.04	1.760 ± 0.01	1.780 ± 0.02	n.a.	n.a.	n.a.
620 nm absorbance	0.430 ± 0.02	0.420 ± 0.02	0.410 ± 0.01	0.420 ± 0.01	n.a.	n.a.	n.a.

**Table 7 foods-14-01961-t007:** Chemical–physical analysis of the 2025 samples.

Parameters Analyzed	Traditional Aging	Underwater Aging
Total acidity (g/L)	5.50 ± 0.02	5.53 ± 0.01
Volatile acidity (g/L)	0.88 ± 0.01	0.87 ± 0.01
Reducing sugars (g/L)	1.00 ± 0.01	1.00 ± 0.01
pH	3.67 ± 0.01	3.67 ± 0.01
Alcohol content (% *v*/*v*)	13.94 ± 0.01	13.94 ± 0.02
Free SO_2_ (mg/L)	21 ± 2	24 ± 2
Total SO_2_ (mg/L)	107 ± 3	111 ± 2
CO_2_ (g/L)	0.07 ± 0.01	0.05 ± 0.01
Dissolved O_2_ (mg/L)	0.086 ± 0.02	0.020 ± 0.03
Total polyphenols (mg/L)	2.239 ± 0.01	2.426 ± 0.02
Total anthocyanins (mg/L)	68 ± 2	75 ± 1
420 nm absorbance	1.760 ± 0.08	1.700 ± 0.10
520 nm absorbance	1.480 ± 0.04	1.450 ± 0.05
620 nm absorbance	0.210 ± 0.02	0.210 ± 0.04

## Data Availability

The original contributions presented in the study are included in the article; further inquiries can be directed to the corresponding author.

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
