# Peer review of "Red Wine Aging Techniques in Spring Water"

_foods, 2025, doi:10.3390/foods14111961_

Round 1
Reviewer 1 Report
Comments and Suggestions for Authors
This paper presents an initial exploration of underwater wine aging, focusing on monitoring various parameters and assessing the impact of this technique. Additionally, it is the first ever study to investigate the use of spring water for wine aging. The findings suggest that underwater aging affects wine quality differently depending on the technique used and the period of time considered. Overall, the article is well organized and its presentation is good. There are several issues in this paper.
(1)The three line table should be used.
(2) Graph and figure. What is the difference? Just use the figure.
(3) The table annotation should be placed before the table.
(4) The experience gained in recent years by various producers who have experimented with this technique highlights a particular influence of the aforementioned environmental parameters on the refinement of some wines and their chemical and organoleptic characteristics. More detailed discussion is needed.
(5) Figure 1-11, 16. Place in the supporting materials.
(6) What is Traditional aging?
(7) Chemical-physical analysis of 2021 samples should be Chemical-physical analysis in 2021?
(8) Figures 12 and 13 highlight this trend at different time scales. What conclusions can be obtained from the figures?
(9) An unpaired two-tailed Student’s test was used to calculate p values of data, and the analysis was performed using XLStat (Figures 14 and 15). Then? What conclusions can be obtained?
(10) As the aging period was extended to 36 months, many of the data differences diminished. Many of the data? Where?
Author Response
Comments 1: The three line table should be used. |
Response 1: Thank you for pointing this out. We agree with this comment. We have corrected the tables 1, 2, 4, 5, 6 and 7. In the revised manuscript the changes can be found on: · page 7, paragraph 2.2.4, and line 227 · page 9, paragraph 2.2.6, and line 294 · page 10, paragraph 3.1, and line 324 · page 12, paragraph 3.2, and line 350 · page 13, paragraph 3.3, and line 377 · page 14, paragraph 3.4, and line 401.
|
Comments 2: Graph and figure. What is the difference? Just use the figure. |
Response 2: Agree. We have, accordingly, changed “Graph” to “Figure”. You can find the changes in the revised manuscript on: · page 15, paragraph 3.4, and line 423 · page 16, paragraph 3.4, and lines 428 and 434 · page 18, paragraph 3.6, and line 487 · page 19, paragraph 3.6, and lines 491 and 498.
|
Comments 3: The table annotation should be placed before the table. |
Response 3: Agree. We have, accordingly, replaced all the table annotations before the tables. In the revised manuscript the changes can be found on: · page 7, paragraph 2.2.4, and line 227 · page 9, paragraph 2.2.6, and line 294 · page 10, paragraph 3.1, and line 324 · page 12, paragraph 3.2, and line 350 · page 13, paragraph 3.3, and line 377 · page 14, paragraph 3.4, and line 401. |
Comments 4: The experience gained in recent years by various producers who have experimented with this technique highlights a particular influence of the aforementioned environmental parameters on the refinement of some wines and their chemical and organoleptic characteristics. More detailed discussion is needed. |
Response 4: Agree. We have, accordingly, revised the text, adding more detailed evaluations to emphasize this point. In fact, as reported in the referenced articles, some results showed that the underwater aging significantly influenced the chemical composition of the wines. The phenolic profile and compounds proved to be the most affected by the two different kinds of aging. The discriminant sensory test also highlighted that the underwater and cellar wines were perceived differently according to the aging treatment. However, other studies suggested that aging wine underwater does not induce significant alterations in its fundamental characteristics compared to traditional cellar aging. In the revised manuscript, the change can be found on page 2, paragraph 1, and line 77. |
Comments 5: Figure 1-11, 16. Place in the supporting materials. |
Response 5: Agree. As suggested, we have added the pictures in the Supporting Materials. In the revised manuscript, this change can be found on page 20 and line 542. |
Comments 6: What is Traditional aging? |
Response 6: As “Traditional aging” we mean the traditional method for aging red wine, based on practices consolidated over the centuries, and still used today in many wineries around the world, especially to produce high-quality wines. This method mainly involves aging in wooden barrels (mainly oak), and the use of winemaking techniques that emphasize the balance between structure, aromas, color, and complexity, as mentioned on page 1, paragraph 1, and line 39.
|
Comments 7: Chemical-physical analysis of 2021 samples should be Chemical-physical analysis in 2021? |
Response 7: In 2021, we carried out the chemical-physical analysis of the wine samples collected in 2021. The same regarding the 2022 and 2023 samples. |
Comments 8: Figures 12 and 13 highlight this trend at different time scales. What conclusions can be obtained from the figures? |
Response 8: Agree. This trend it confirms the tendency to reduce the variability of temperature and relative humidity values, thanks to the confined and controlled environment of the cellar, with reduced daily and seasonal fluctuations. Figures 12 and 13, considering the dataset both as a whole (from June 2020 to February 2025) and at the level of detail of the single month, confirm the validity of the initial assumption of investigating the aging of wine in a particular environment such as a well. The mentioned figures can be found in the revised manuscript on page 11, paragraph 3.1, and lines 307 and 311. |
Comments 9: An unpaired two-tailed Student’s test was used to calculate p values of data, and the analysis was performed using XLStat (Figures 14 and 15). Then? What conclusions can be obtained? |
Response 9: Agree. We have, as suggested by Reviewer 2, modified the statistical analysis, also to emphasize this point. A one-way ANOVA was used to compare the means of two groups to see if there are any significant differences (p<0.5), always using XLStat software. The analysis showed that the p-value obtained is lower than the significance level (p=0.05), and therefore there is a significant difference between the means of temperatures values detected by the two probes, especially in the years 2021 and 2022, while in the first year of testing (2020), the values found were significantly different only for one month (November with a p=0.007). In the first case (2021) the months in which the differences were found were August (p=0,028), November (p<0.0001), and December (p<0.0001). Instead, for 2022 the differences between the two groups of temperature values affected 4 months out of 7 and more precisely June (p=0.013), August (p=0.049), October (p<0.0001), and December (p=0.001). Beyond the differences found with the statistical analysis, the values recorded with the two probes highlighted how the two types of conservation environments compared (cellar and well) were characterized by temperatures that were not very different from each other. The differences in terms of degrees were revealed to be a little more evident (by a few degrees) especially in the winter months in which the degrees measured in the water were 1-2 values higher than those in the cellar, confirming the property of water to maintain a constant temperature despite the fluctuations of the external environment. In the revised manuscript, the changes can be found on page 9, paragraph 3.1, and line 297.
|
Comments 10: As the aging period was extended to 36 months, many of the data differences diminished. Many of the data? Where? |
Response 10: Agree. We have, accordingly, revised the text, adding more detailed evaluations to emphasize this point. Precisely the parameters that show the greatest difference in the behavior of the wine between the two aging systems are those that show the least differences in the last two years of experimentation. In particular, dissolved oxygen differs after the first year by 0.739 mg/L (0.763 mg/L and 0.024 mg/L), while in the second year by 0.112 mg/L (0.143 mg/L and 0.031 mg/L) and in the third year by 0.028 mg/L (0.042 mg/L and 0.014 mg/L). The behavior of total anthocyanins also shows minor differences between the two aging methods in the last two years of experimentation. In more detail, they differ after the first year by 68 mg/L (95 mg/L and 163 mg/L), while in the second year by 1 mg/L (47 mg/L and 46 mg/L) and in the third year by 5 mg/L (54 mg/L and 59 mg/L). In the revised manuscript the changes can be found on: · page 15, paragraph 3.4, and line 408.
|
Reviewer 2 Report
Comments and Suggestions for Authors
The authors of the manuscript "Red wine aging techniques in spring water" conducted research related to the parameters of wine aging in various conditions, with the main emphasis on the aging processes of red wine in spring water. The methodology and layout of the work are correct, but in order to clarify certain methodological and research issues, there is a need to correct the manuscript. In the discussion, references to the results of other studies related to the aging processes of wines are too poor. Other more detailed comments are given in the attached file.

Author Response
Comments 1: Describe the oenological procedures used in the production of the wine being tested. |
Response 1: Thank you for pointing this out. We agree with this comment. We have added a reference related to the production regulations of this wine. In the revised manuscript the changes can be found on: · page 3, paragraph 2.2, and line 115.
|
Comments 2: Briefly describe the research methodology used and the equipment used and its operating parameters. |
Response 2: Agree. We have, accordingly, added more detailed description of the research methodology used, adding the indication of the methods used to determine polyphenols and anthocyanins. The determination of the total polyphenol content was carried out using the Folin-Ciocalteu method, measuring the absorbance of a chemically treated sample with a spectrophotometer. The identification of the total anthocyanins content was carried out using the Ribéreau-Gayon and Stonestreet method, measuring the absorbance of a chemically treated sample with a spectrophotometer. In the revised manuscript, the changes can be found on page 8, paragraph 2.2.4, and line 239.
|
Comments 3: Please complete the results given in tables 2, 3 and 4 with the standard deviation. In addition, in the table caption write that the results are given as an average of three measurements. |
Response 3: Agree. We have, accordingly, completed all the table with the standard deviation values. In the revised manuscript the changes can be found on: · page 9, paragraph 2.2.6, and line 294 · page 10, paragraph 3.1, and line 324 · page 12, paragraph 3.2, and line 350 · page 13, paragraph 3.3, and line 377 · page 14, paragraph 3.4, and line 401 · page 18, paragraph 3.6, and line 484. |
Comments 4: Replace commas with periods in all tables. |
Response 4: Agree. As suggested we have replaced commas with periods in all tables. These changes can be found in the revised manuscript on: · page 9, paragraph 2.2.6, and line 294 · page 10, paragraph 3.1, and line 324 · page 12, paragraph 3.2, and line 350 · page 13, paragraph 3.3, and line 377 · page 14, paragraph 3.4, and line 401 · page 18, paragraph 3.6, and line 484. |
Comments 5: Explain the abbreviation used abs. |
Response 5: The abbreviation “abs” means “absorbance”. Following your comment, we have removed them from all tables, since absorbance is a dimensionless derived unit and it does not appear in the International System of Units. These changes can be found in the revised manuscript on: · page 12, paragraph 3.2, and line 350 · page 13, paragraph 3.3, and line 377 · page 14, paragraph 3.4, and line 401 · page 18, paragraph 3.6, and line 484. |
Comments 6: Supplement the result with the standard error. |
Response 6: We have, accordingly, completed the suggested table with the standard deviation values. In the revised manuscript the changes can be found on: · page 9, paragraph 2.2.6, and line 294.
|
Comments 7: The abbreviations used are described in the figure caption. |
Response 7: Agree. We have, accordingly, modified the figure caption to emphasize this point. In the revised manuscript the change can be found on page 10, paragraph 3.1, and line 311. |
Comments 8: With such complex and numerous data, the statistics used seem to be not very graphic. I recommend performing more advanced statistical analyses that allow for a more comprehensive analysis of the research results, I mean ANOVA type analyses and factor analysis or heat maps, which will allow for a more reliable interpretation of the obtained results. |
Response 8: Agree. We have, accordingly, modified the statistical analysis, in order to emphasize this point. A one-way ANOVA was used to compare the means of two groups to see if there are any significant differences (p<0.5), always using XLStat software. Furthermore, to have a better graphical result, we have performed a comparative heat map and added a table with all the statistical results. In the revised manuscript the change can be found on page 9, paragraph 3.1, and line 314. |
Comments 9: In the graphic caption, please add the methods used to determine polyphenols. |
Response 9: Agree. We have, accordingly, modified the graphic caption, adding the indication of the methods used to determine polyphenols. The determination of the total polyphenol content was carried out using the Folin-Ciocalteu method, measuring the absorbance of a chemically treated sample with a spectrophotometer. In the revised manuscript, the change can be found on page 16, paragraph 3.4, and line 428. |
Comments 10: In the caption of the graphic, please add the methods used to identify the anthocyanins. |
Response 10: Agree. We have, accordingly, modified the graphic caption, adding the indication of the methods used to determine anthocyanins. The identification of the total anthocyanins content was carried out using the Ribéreau-Gayon and Stonestreet method, measuring the absorbance of a chemically treated sample with a spectrophotometer. In the revised manuscript, the change can be found on page 16, paragraph 3.4, and line 434. |
Comments 11: similar to graphic no. 2. |
Response 11: Agree. We have, accordingly, modified the graphic caption, adding the indication of the methods used to determine polyphenols. The determination of the total polyphenol content was carried out using the Folin-Ciocalteu method, measuring the absorbance of a chemically treated sample with a spectrophotometer. In the revised manuscript, the change can be found on page 19, paragraph 3.6, and line 491. |
Comments 12: similar to graphic no. 3. |
Response 12: Agree. We have, accordingly, modified the graphic caption, adding the indication of the methods used to determine anthocyanins. The identification of the total anthocyanins content was carried out using the Ribéreau-Gayon and Stonestreet method, measuring the absorbance of a chemically treated sample with a spectrophotometer. In the revised manuscript, the change can be found on page 19, paragraph 3.6, and line 498. |
Comments 13: The discussion contains too few references to the results of other studies on wine ageing techniques. |
Response 13: Agree. We have, accordingly, modified the text to emphasize this point. More in detail, we have added the reference to some further study related to red wine aging in different conditions. In the revised manuscript, the change can be found on page 13, paragraph 3.2, and line 358, and on page 20, paragraph 4, and line 528.
|
Reviewer 3 Report
Comments and Suggestions for Authors
The topic of wine aging under spring water is creative and the authors presented an initial exploration in this area; however, the manuscript requires major revision.
- The Materials and Methods section lacks clarity. It is not easy to understand. Consider moving the results of the chemical-physical analysis for the 2021, 2022, and 2023 samples to the Results section.
- The manuscript did not include any statistical analysis. Without it, the study is not suitable for publication in a scientific journal.
- Make sure consistency in the number of decimal places used for each parameter.
- Combine the “Parameters Analyzed” and “Unit of Measurement” into a single column. Also, change the unit “l” to uppercase “L”.
- Replace the term “Graph” with “Figure.”
- Use figures to present the organoleptic analysis results. Additionally, specify how many panelists participated in the evaluation.
- For practical application, clarify how many bottles can be placed in a well. Note that wineries typically do not have many wells, so this should be explained more realistically.
Author Response
Comments 1: The Materials and Methods section lacks clarity. It is not easy to understand. Consider moving the results of the chemical-physical analysis for the 2021, 2022, and 2023 samples to the Results section. |
Response 1: Thank you for pointing this out. We agree with this comment. We have made some changes to the text of the Materials and Methods section and we have moved the tables 3, 4, and 5, in order to make the manuscript clearer and more understandable, in addition to the numerous photographs reported to describe the experimental conditions. In the revised manuscript the changes can be found on: · page 12, paragraph 3.2, and line 350 · page 13, paragraph 3.3, and line 377 · page 14, paragraph 3.4, and line 401.
|
Comments 2: The manuscript did not include any statistical analysis. Without it, the study is not suitable for publication in a scientific journal. |
Response 2: Agree. We have added a statistical analysis, to emphasize this point. A one-way ANOVA was used to compare the means of two groups to see if there are any significant differences (p<0.5), always using XLStat software. The analysis showed that the p-value obtained is lower than the significance level (p=0.05), and therefore there is a significant difference between the means of temperatures values detected by the two probes, especially in the years 2021 and 2022, while in the first year of testing (2020), the values found were significantly different only for one month (November with a p=0.007). In the first case (2021) the months in which the differences were found were August (p=0,028), November (p<0.0001), and December (p<0.0001). Instead, for 2022 the differences between the two groups of temperature values affected 4 months out of 7 and more precisely June (p=0.013), August (p=0.049), October (p<0.0001), and December (p=0.001). Beyond the differences found with the statistical analysis, the values recorded with the two probes highlighted how the two types of conservation environments compared (cellar and well) were characterized by temperatures that were not very different from each other. The differences in terms of degrees were revealed to be a little more evident (by a few degrees) especially in the winter months in which the degrees measured in the water were 1-2 values higher than those in the cellar, confirming the property of water to maintain a constant temperature despite the fluctuations of the external environment. In the revised manuscript, the changes can be found on page 9, paragraph 3.1, and line 297.
|
Comments 3: Make sure consistency in the number of decimal places used for each parameter. |
Response 3: Agree. We have, accordingly, verified and in some cases modified the number of decimal places used for each parameter. In the revised manuscript the changes can be found on: · page 12, paragraph 3.2, and line 350 · page 13, paragraph 3.3, and line 377 · page 14, paragraph 3.4, and line 401. |
Comments 4: Combine the “Parameters Analyzed” and “Unit of Measurement” into a single column. Also, change the unit “l” to uppercase “L”. |
Response 4: Agree. We have, accordingly, revised the text, following the suggested changes. In the revised manuscript, the change can be found on: · page 10, paragraph 3.1, and line 324 · page 12, paragraph 3.2, and line 350 · page 13, paragraph 3.3, and line 377 · page 14, paragraph 3.4, and line 401 · page 18, paragraph 3.6, and line 484. |
Comments 5: Replace the term “Graph” with “Figure.” |
Response 5: Agree. We have, accordingly, changed “Graph” to “Figure”. You can find the changes in the revised manuscript on: · page 15, paragraph 3.4, and line 423 · page 16, paragraph 3.4, and lines 428 and 434 · page 18, paragraph 3.6, and line 487 · page 19, paragraph 3.6, and lines 491 and 498. |
Comments 6: Use figures to present the organoleptic analysis results. Additionally, specify how many panelists participated in the evaluation. |
Response 6: As suggested, we added more information related to the organoleptic analysis. Since the rate used for grades of wine during sensory evaluation was a unipolar scale (verbal categories), we don't know what kind of figures we can use to better clarify the results. The revised manuscript now have in the section 2.2 a new subsection entitled Organoleptic analysis. In this subsection authors gave more information regarding technical panel involved in sensory analysis, number of members in this panel (25), their expertise, how long they are involved in the sensory analysis of wine and which kind of rate was used for grades of wine during sensory evaluation. In the revised manuscript, the changes can be found on page 8, paragraph 2.2.5, and line 254.
|
Comments 7: For practical application, clarify how many bottles can be placed in a well. Note that wineries typically do not have many wells, so this should be explained more realistically. |
Response 7: In the specific case under examination, up to 222 bottles of wine were stored in the well. The size of this well can certainly accommodate up to 500 bottles. In the Piedmont region, many wineries and agricultural companies have a well with spring water. However, the characteristic of the individual well is not important for the purposes of our study, but rather the fact of experimenting with a technique for aging red wine in spring water. |
Round 2
Reviewer 1 Report
Comments and Suggestions for Authors
Accept
Author Response
Thank you very much for taking the time to review this manuscript. I think I don't have new revisions to submit.